# The Effects of a Comprehensive, Integrated Obesity Prevention Intervention Approach (SuperFIT) on Children’s Physical Activity, Sedentary Behavior, and BMI Z-Score

**DOI:** 10.3390/ijerph16245016

**Published:** 2019-12-10

**Authors:** Ilona van de Kolk, Sanne M. P. L. Gerards, Lisa S. E. Harms, Stef P. J. Kremers, Jessica S. Gubbels

**Affiliations:** Department of Health Promotion, NUTRIM School for Nutrition and Translational Research in Metabolism, Maastricht University, P.O. Box 616, 6200 MD Maastricht, The Netherlands; sanne.gerards@maastrichtuniversity.nl (S.M.P.L.G.); lisa.harms@maastrichtuniversity.nl (L.S.E.H.); s.kremers@maastrichtuniversity.nl (S.P.J.K.); jessica.gubbels@maastrichtuniversity.nl (J.S.G.)

**Keywords:** BMI z-score, children, home, intervention, physical activity, preschool, sedentary behavior

## Abstract

SuperFIT is a comprehensive, integrated intervention approach aimed at promoting healthy energy balance-related behaviors in 2- to 4-year-old children in the preschool and home settings. A quasi-experimental research design was adopted to evaluate the effects of SuperFIT on physical activity (PA), sedentary behavior (SB) and Body Mass Index (BMI) z-score. Children could participate in the preschool-based and family-based component (full intervention) or only in the preschool-based component (partial intervention). Children’s PA levels and SB were assessed with accelerometers and observations, and height and weight were measured for the BMI z-score. Measurements were performed at baseline and two follow-up time points. Effectiveness was evaluated using linear mixed-model analyses, correcting for relevant covariates. Healthy changes in PA levels occurred within all study groups over time. No significant differences were found in overall PA levels between the intervention groups and control group at both follow-ups. Nevertheless, sedentary behavior decreased more in the full intervention group (effect size (ES): −0.62), and moderate-to-vigorous PA (ES: 0.85) and counts per minute (ES: 0.45) increased more compared to the control group on preschool days at the first follow-up. No effects were found for BMI z-score. The integrated approach of SuperFIT may induce changes in PA of young children, although the effects were small.

## 1. Introduction

Regular and sufficient physical activity (PA) is an important contributor to the physical health and psychosocial well-being of children [1,2]. Early childhood is an important period for developing healthy habits, such as participating in PA, and PA habits are known to track from childhood into adulthood [3,4]. PA at a young age is also essential for the development of fundamental motor skills, which in turn is predictive for PA at an older age [5]. However, research has shown that young children are not getting enough daily PA, plus their daily movement patterns are characterized by large amounts of sedentariness [6,7,8,9].

A lack of PA in combination with high sedentariness and unhealthy nutrition is associated with childhood overweight and obesity [10,11]. These are continuing important public health problems with a prevalence that is expected to rise even further [12]. In the Netherlands, around 8% of 2-year-old children are overweight or obese and this prevalence increases with age [13]. Overweight and obesity are related to various health issues in both childhood and adulthood [14,15]. Promoting PA in young children is therefore crucial to supporting children’s healthy lifestyles and health.

The home setting exerts an important influence on the behavior of young children. Not only is parental PA behavior related to child PA [16], parental support and family characteristics are also related to PA [17]. The use of supportive parenting practices may promote PA among children [18]. In addition, many young children are enrolled in Early Care and Education (ECE), which increases with age (from ~33% for 0–2-year-olds to ~80% for 3-year-olds) [19]. The ECE setting has been both positively and negatively associated with PA in young children in different studies [20,21]. In general, children spend much time sedentary and little time in PA in the ECE setting [22]. Educator-related factors (e.g., activity-related practices and presence), physical environment-related factors (e.g., outdoor play area and larger play spaces), and organization-related factors (e.g., provision of active opportunities) influence PA within the ECE setting [23]. Other factors are thought to be related to children’s PA, although the evidence is less conclusive, such as the availability of portable or fixed play materials, educator training, and indoor play space [23]. The ECE setting is considered an essential and promising one for interventions in order to improve children’s PA [24]. 

Systems theory suggests that it is important to take into account both the ECE and home settings when intervening to promote healthy behavior [25,26]. Furthermore, aligning important micro-settings towards more supportive environments for healthy behavior may result in synergistic effects [27]. Research has already shown that this combination of ECE and home is more effective in preventing childhood overweight and obesity than targeting just one setting, although the relationship with behavior is less clear [28,29]. On the other hand, it has been found that inconsistencies between the ECE and home settings are related to less physical activity in young children [30]. Including the community setting and creating a health-supporting community may also support intervention effectiveness as this may foster sustainable change [31].

Interventions tend to focus on single settings or single aspects of childhood overweight and obesity and as a result may lack comprehensiveness. Therefore, SuperFIT was developed as a comprehensive, integrated intervention approach in the ECE and home settings in the Netherlands. It takes into account the interaction between settings and the complexity of the childhood overweight and obesity problem. It primarily aims to increase physical activity levels, decrease sedentary behavior and increase healthy nutrition behavior (e.g., drinking water and eating fruits and vegetables) of young children through changes in the sociocultural environment (i.e., use of supportive physical activity and nutrition-related practices by preschool teachers and parents) and physical environment (i.e., availability of space, play materials, healthy foods). Through these behavioral changes, SuperFIT also aims to prevent childhood overweight and obesity. The current study evaluated the effectiveness of SuperFIT on child physical activity, sedentary behavior and Body Mass Index (BMI) z-score. We hypothesize that SuperFIT will increase the levels of physical activity and decrease the levels of sedentary behavior in young children compared to the control group. In addition, we hypothesize that SuperFIT will help children maintain or achieve a healthy BMI z-score. Finally, we hypothesize that the combination of the preschool and home setting will lead to synergy, increasing intervention effectiveness.

## 2. Materials and Methods

A protocol with a detailed description of the SuperFIT intervention and evaluation can be found elsewhere [32].

### 2.1. Study Design

In the Netherlands, ECE consists of two forms. In one, day-care centres provide full-day childcare [33], which children aged 0- to 4-years old can attend. In another, preschools provide half-day childcare with a specific goal to prepare children in a playful way for primary school [33]. Children aged 2- to 4-years-old can attend preschools. Parents can receive a general childcare benefit for both of these forms of ECE from the government, based on their working hours and income [34]. ECE centres have a large reach as children with, for example, language or socio-emotional developmental delays can be referred to ECE to enter a program to catch up on these delays [35]. SuperFIT was implemented at preschools and was evaluated using a quasi-experimental research design.

### 2.2. The Intervention

SuperFIT is a comprehensive, integrated intervention approach based on socio-ecological models of behavior and systems theory [25,27]. In particular, it aims to align intervention strategies between different micro-settings (i.e., preschool and home) to enhance intervention effectiveness. SuperFIT was developed in partnership with the intervention preschool organization, a local PA providing organization, and health promotion experts. Formative research [36], practice-based knowledge of cooperating partners, and theory- and evidence-based knowledge [26,37,38,39,40] were used to develop it. Further, a continuous process of co-creation, feedback and adaptations was adopted during development to increase suitability and applicability in the settings. The intervention consisted of preschool-based, family-based, and community components. 

Within the preschool-based component, several strategies were implemented that targeted the sociocultural and physical environment [41]. The sociocultural environment was operationalized as the physical activity- and nutrition-related practices of the preschool teachers. The intervention strategies involved the preschool teachers and included (1) an inspirational session with a school-based PA expert, (2) three off-the-job interactive training sessions on PA, nutrition, and positive child rearing, led by an expert on the specific topic (3) on-the-job coaching sessions following each the off-the-job training (three in total), on PA or nutrition led by the same experts, and (4) cards with easy-to-perform PA games and nutrition-related activities to support preschool teachers to integrate PA and healthy nutrition in the curriculum. The physical environment was operationalized as all that is tangibly available for the children at the preschool. Intervention strategies were (1) provision of a box of general play materials aligned with the PA-related cards that could promote PA both indoors and outdoors (e.g., bean bags, hoops, balls); (2) general nutrition-related materials aligned with the nutrition-related cards (e.g., water tap, fruit and vegetable toys, nutrition-related story books); and (3) complementary fruit and vegetables delivery. A local greengrocer supplied less-familiar fruits or vegetables (e.g., avocado, raspberries, carrots in different colors) to increase variety. These supplemental fruit and vegetables were available every day and were similar every two weeks to ensure repeated exposure. Intervention strategies were mostly designed to serve as add-in rather than add-on activities, which were highly adaptable to the specific situation of a preschool, to support implementation and sustainability.

The family-based component was developed to provide fun activities to help families integrate PA and healthy nutrition in their daily life [42]. All possible caregivers were invited to participate in these sessions. Caregiver-only sessions were organized in order to address the influences of the different types of environments (i.e., sociocultural, physical, political and economic [41]) on nutrition and PA. These sessions were based on Lifestyle Triple P seminars and were given by a trained Triple P implementer [43]. Three Triple P seminars were provided (one on PA, one on nutrition and one on general parenting), which lasted around 1.5 h. During the caregiver-only sessions PA or nutrition-related activities were organized for the children and childcare was available for the youngest siblings. In addition, family sessions were organized. They were characterized by fun activities for the whole family. The PA-related family sessions aimed at co-physical activity, performing active games that are easily translated to the home setting. The nutrition-related family sessions consisted of, for example, taste sessions, and making healthy treats. 

The community component aimed to increase linkages between different organizations involved in young children’s physical activity and nutrition behavior. A social map was distributed that indicated PA opportunities suitable for young children within the community. Intervention activities in both the preschool-based and family-based components took place between April 2017 and May 2018. Activities in the community component started during the same period, but the social map was distributed in June 2018.

### 2.3. Study Population and Recruitment

A convenience sample of intervention preschools was recruited by a childcare organization in an urban municipality in Limburg (the Netherlands). Preschools were selected based on the socio-economic status (SES) of their neighborhood as determined by the 2014 values of the Netherlands Institute for Social Research (SCP) [44]. A negative neighborhood SES score indicated low SES, and preschools in these neighborhoods were eligible. In consultation with the childcare manager a selection of eligible preschools was determined. In total, twelve intervention preschools participated. Control preschools were selected in another urban municipality in Limburg (the Netherlands) by one childcare organization. Preschools in neighborhoods with comparable SES scores were eligible [44]. Nine preschools participated. Due to the nature of the project, no randomization or blinding was performed. Children attending the participating preschools were eligible for participation in the research of SuperFIT. Additional inclusion criteria for the children were: (1) At least one parent had to be able to understand Dutch; and (2) both parents signed informed consent forms. Parents were provided with the choice to participate in either just the preschool-based component (partial intervention) or both the preschool-based and family-based components (full intervention). All preschool teachers working at the participating preschools were eligible to participate in the research after providing informed consent. All children attending the participating preschools were exposed to SuperFIT, although not all children participated in the research. All preschool teachers were expected to participate in the intervention training and coaching sessions by the childcare organization as part of their professional development. The Maastricht University Medical Centre+ Medical Ethics Committee reviewed and approved this study (METC163022/NL 58061.068.16), and the trial was prospectively registered (Clinicaltrials.gov, NCT03021980). 

### 2.4. Measurements

Baseline measurements were performed from January until July 2017. Follow-up measurements took place in November/December 2017 (first follow-up) and May/June 2018 (final follow-up). In order to reduce the participant burden, data collection was aligned with intervention participation. This means that more extensive data collection was done for participants in the family-based component compared to those in the preschool-based component or control group. Measurements were performed on anthropometry, physical activity, dietary intake and preschool teacher and parent nutritional and physical activity-related practices. The current paper presents the effects on BMI z-score and physical activity. The effects of SuperFIT on dietary intake will be presented elsewhere [45].

#### 2.4.1. Physical Activity

Children’s PA was assessed using Actigraph GT3X+ (Actigraph, Pensacola, FL, USA) accelerometers, applying an adjusted wearing protocol [46]. Accelerometers were placed on the right hip using an elastic belt. Children wore the accelerometers for eight consecutive days during waking hours, excluding activities involving water such as bathing/showering and swimming. Data were derived using a 10-s epoch. Wear time validation by Troiano (2007) was used [47]. Minimal wear time was set at 360 min per day. For children whose wear time indicated the accelerometer was worn during night sleep, data were extracted from 6.00 am to 9.00 pm to exclude sleep time. Children who had one day of sufficient wear time were included in the analysis. Uniaxial cut-off points of Pate et al. (2006) for PA intensity were used [48]. In addition, time spent in the different PA categories (sedentary behavior (SB), light physical activity (LPA) and moderate-to-vigorous physical activity (MVPA)) was divided by total wear time to calculate the percentage of time spent in each category. Counts per minute (CPM) based on vector magnitude were extracted. Data were extracted for overall PA and PA on preschool days. 

#### 2.4.2. Observations at Preschools during Implementation

Observations were performed at a random selection of the intervention preschools (9 out of 12 preschools, 10 groups) to assess change in daily activities and the preschool environment, among other things. The observations took place during morning opening hours on one day. They were performed twice during implementation (September/October 2017 and April 2017) and once after implementation (September/October 2018). The observations were performed using an observation form based on the Environment and Policy Assessment and Observation form (EPAO) [49]. The observation form consisted of the parts related to daily activities, the social and physical environments, and was adjusted to the Dutch preschool setting. The form followed the structure of a regular preschool day, starting with activities before snack time (indoor and outdoor), snack time, and activities after snack time (indoor and outdoor). For each activity, duration was measured by recording start and finish time. Activities outside, activities initiated by preschool staff both inside and outside (e.g., throwing over a ball), sedentary activities both inside and outside (e.g., doing handcrafts or playing seated in the sandpit), and circle time could be recorded separately. Observations were performed on a group level, indicating that activities should involve the majority of the children present to be recorded.

#### 2.4.3. Anthropometrics

A trained member of the research team assessed the children’s weight and length, using a standardized protocol. Standing height was measured using the Seca © 213 stadiometer (Seca, Hamburg, Germany), without shoes, to the nearest decimal in centimeters (cm). Weight was measured using the Seca © Clara 803 digital weighing scale to the nearest decimal in kilograms (kg). Heavy clothing and shoes were removed before measurement. Particular events occurring during the measurements, such as not wanting to take shoes off or wearing heavy clothing, were recorded to aid data cleaning. Further data cleaning was done on data entry errors and outliers. Height and weight measurements were used to calculate BMI, which was converted to a BMI z-score, adjusted for age, gender, and ethnicity, using a Dutch reference population (The Fifth Dutch Growth Study) [13].

#### 2.4.4. Covariates

A baseline parental questionnaire was used to measure a range of demographics. For the children, these included birthdate and gender. Child birthdate was used to calculate the child’s age at baseline. Parental demographics included parental birthdate, education level and country of birth. Parental birthdate was used to calculate parental age at baseline. Education level was recoded into low, medium and high using the International Standard Classification of Education (ISCED) 2011 classification [50]. Country of birth was recoded into ‘the Netherlands’ or ‘other’. The questionnaire was also used to measure parental weight and length in order to calculate BMI.

To be able to correct for weather influences, data on weather conditions between 6 a.m. and 11 p.m. were collected from the Royal Dutch Meteorological Institute (KNMI) [51]. Data were gathered on temperature (average degree Celsius), sunshine (total hours) and precipitation (total hours).

### 2.5. Statistical Analyses

Baseline characteristics and outcome values were analyzed for differences between the groups using ANOVA for continuous variables and chi-square tests for categorical variables. Effects of SuperFIT on overall PA, PA on preschool days, and BMI z-score were analyzed using multiple linear mixed models with child and preschool levels in order to correct for repeated measurements and group effects. Linear mixed models handle missing outcome values by imputing them with a likelihood-based method. The fixed part of the model consisted of condition, time and the interaction term of condition and time. A random intercept was included when this significantly improved the model based on the likelihood ratio. Sensitivity analysis was performed using only those cases that provided data for all measurements. PA outcomes were corrected for child age at baseline, child gender and weather conditions (temperature, sunshine, and precipitation). BMI z-score was adjusted for parental BMI at baseline, parental education level, parental country of birth. Manual backwards analysis was applied to correct for possible covariates.

Descriptive analyses were performed on the data from the observations. Total minutes spent in each activity were calculated. All analyses were performed using SPSS version 25.0 (IBM Corp, Armonk, NY, USA), and *p*-values < 0.05 were considered statistically significant. For all PA outcomes and BMI z-scores, effect sizes were calculated based on the estimated mean difference between measurements and the standard errors of the estimated means. Cohen’s classification was used to determine the level of effect size [52].

## 3. Results

### 3.1. Participants

A total of 191 children participated in the study at baseline. Forty-seven children were included in the full intervention, 52 in the partial intervention and 92 in the control group. The children were 3.1 years old on average, and 46.1% were boys (see Table 1). At baseline, the groups differed significantly by parental country of birth, parent education level, and partner education level. 

Valid accelerometer data for at least one measurement were available for 175 children (91.6%). Valid accelerometer data were available for 143 children (74.9%) at baseline, 129 children (67.5%) at the first follow-up, and 120 children (62.8%) at the final follow-up. Anthropometric measurements were available for 180 children (94.2%) for at least one measurement (baseline 164 children (85.9%), first follow-up measurement 146 children (76.4%), and final follow-up measurement 136 children (71.2%)). At baseline, parents of 127 children (66.5%) filled out the parental questionnaires.

### 3.2. Physical Activity at Preschools

The observation data showed that during the implementation period of SuperFIT, preschools reduced time spent sedentary, increased time spent active, and increased time spent outdoors (Table 2). Although average time spent outdoors increased, some preschools did not go outside at all during the observation days. Furthermore, the average duration of circle time and snack time appeared rather consistent. However, the maximum time spent in circle time decreased substantially.

### 3.3. Effects on Children’s Physical Activity Outcomes

There were no significant differences for the PA outcomes between the groups at baseline. Regarding children’s PA on preschool days, the results are shown in Table 3. Children in the full intervention showed significant within-group differences at both the first (SB and MVPA) and the final follow-up (all PA outcomes). In addition, compared to the control group, the full intervention showed significant differences for SB (effect size: −0.62), MVPA (effect size: 0.85), and CPM (effect size: 0.45). These significant differences were not present at the final follow-up.

Children in the partial intervention and the control group showed significant within-group differences between baseline and final follow-up, but not for the first follow-up. No significant differences were seen between the partial intervention and control group for both the first and final follow-up for PA outcomes on preschool days. Based on the effect sizes, the control group improved more on the PA outcomes than the partial intervention group, except for CPM at the first follow-up and MVPA at the final follow-up. However, all effect sizes were small or very small.

Table 4 shows the effects of SuperFIT on overall PA outcomes for the three study groups. All groups improved in the PA outcomes at the follow-up measurements. Significant within-group differences were seen between baseline and the final follow-up measurement in all groups. Only light PA was borderline significant (*p* = 0.051) in the full intervention group. At the first follow-up, the full intervention improved more on SB and MVPA compared to the control group. The partial intervention group improved more on MVPA compared to the control group. However, the effect sizes were small or very small and not significant.

For the remaining outcomes, the control group improved more than both the full intervention and partial intervention group. These effect sizes were also small or very small and not significant. The sensitivity analyses showed different results, but comparable conclusions (Appendix A). Effect sizes were small or very small, and no significant differences were found.

### 3.4. Effects on BMI Z-Score

No significant differences in BMI z-score were seen, with very small effect sizes between the full intervention and control group at the first follow-up (observed mean ± SD 0.20 ± 0.98 and 0.13 ± 1.00 resp.; B = −0.09, 95% CI −0.31; 0.13, *p* = 0.44, ES −0.09) and the final follow-up (observed mean ± SD 0.28 ± 0.90 and 0.08 ± 0.94 resp.; B = 0.00, 95% CI −0.25; 0.25, *p* = 0.99 ES 0.01). In addition, no significant differences were seen between the partial intervention and control group on BMI z-score at the first follow-up (observed mean ± SD 0.20 ± 0.81 and 0.13 ± 1.00 resp.; B = 0.05, 95% CI −0.17; 0.26, *p* = 0.66, ES 0.06) and the final follow-up (observed mean ± SD −0.01 ± 0.77 and 0.08 ± 0.90 resp.; B = −0.13, 95% CI −0.38; 0.11, *p* = 0.28, ES −0.14). Effect sizes were also very small. BMI z-score within the partial intervention group improved significantly between baseline and final follow-up measurement (*p* = 0.019). Sensitivity analysis showed similar results and conclusions (Appendix A).

## 4. Discussion

The aim of this study was to evaluate the effectiveness of SuperFIT on child PA and BMI z-scores. Observation data showed that time spent active (inside and outside) in the preschool setting increased during implementation, while time spent sedentary decreased. For PA on preschool days, significant differences were seen on SB, MVPA and CPM between the full intervention group and the control group at the first follow-up. No significant differences were seen between the partial intervention group and the control group, although the partial intervention group improved more on MVPA at the final follow-up. With regard to overall PA, all study groups improved significantly between baseline and final follow-up. Both intervention groups showed a greater improvement on MVPA at the first follow-up than the control group. The full intervention also improved more on SB at the first follow-up compared to the control group. However, these were small or very small effects, and there were no significant differences between the groups. For BMI z-score, no significant changes were seen between both intervention groups and the control group. The partial intervention group showed a significant decrease in BMI z-score between baseline and final follow-up. No significant changes in BMI z-score over time were seen in the full intervention group. 

This limited effectiveness of SuperFIT may be explained by the long causal chain between the intervention and the outcomes assessed. The intervention aimed to change the sociocultural (i.e., the behavior of intermediaries) and the physical environment at home and in the ECE setting. It was hypothesized that these changes would lead to changes in child behavior, which would eventually lead to changes in BMI z-score. It may be that the changes caused by SuperFIT were not substantial enough or the follow-up time was not long enough to result in effects going up the causal chain. It would be interesting to look at the effect of SuperFIT on the intermediate outcomes (e.g., teacher practices), which are closer to the intervention activities. Nonetheless, the observations showed that changes in daily activities occurred at the preschools. A reason why this was only reflected to a limited extent in overall PA outcomes could be that activity time in childcare may be compensated by increased sedentary time at home [21]. The small differences in activity time at the preschools may therefore not be enough to overcome such potential compensation. This may explain, in particular, the lack of effects in the partial intervention group. Although a clear beneficial relationship between PA and health outcomes is established, it remains uncertain what duration of PA is necessary to cause these health benefits, especially in young children, even though more PA may exert greater health benefits [1]. As a result of this lack of evidence, the Netherlands has not yet adopted a PA guideline for children under the age of four [53]. Therefore, it is also difficult to formulate recommendations on the minimum amount of physical activity that should be provided in both the ECE and home settings.

The limited effects found for SuperFIT reflect the outcomes in comparable studies [54,55]. In general, the results of other interventions on PA outcomes have been mixed: Some studies showed effects [56,57], while others showed no effect [58]. In our study, the full intervention seemed to be beneficial for PA on preschool days. This may imply that children benefit more from changes in the preschool setting if there are also changes in the home setting. This underlines the importance of including a family component in preschool-based interventions, which has been suggested previously [29]. This supports the hypothesized synergistic effects of aligning both settings to be more health supportive, although this only occurred in the preschool setting [27]. For all groups, a within-group change of PA was seen. This may be explained by the natural development of PA throughout childhood, with PA levels generally rising until the age of five and then starting to decline [59,60].

Studies have shown the importance of determinants in different types of environments on physical activity and sedentary behavior [36,61,62]. Systematic reviews have also stressed the importance of integrated interventions, i.e., combining the physical and sociocultural environment [63,64]. The integrated approach taken in SuperFIT, in which these different types of environments were taken into account was expected to be supportive for intervention effectiveness. The strategies that were implemented to change the different environments were evidence-based [43,65,66,67]. In addition, a review has shown that environmental changes are most effective to stimulate pre-schoolers physical activity [24]. Due to the comprehensive, integrated approach of SuperFIT, it is difficult to identify which elements supported effectiveness. To increase our understanding, separate evaluations will be performed to study the effects of SuperFIT on the physical and social environment.

Only a few comparable interventions have been shown to be effective on weight-related outcomes [68,69,70,71]. It appears unlikely to find changes (especially significant ones) in weight-related outcomes within the relatively short follow-up periods usually applied in intervention studies. The small effect sizes found in this study, although not consistent over time, are comparable to ones mostly found in preschool-based or school-based interventions [72,73,74]. These small effects may be relevant, in particular, if they are sustained in the long-term as they can prevent overweight and obesity in adulthood [74]. Longer follow-up periods may be needed to be able to show effects on weight-related outcomes. Effectiveness studies are usually performed within time-limited projects; there are practical impossibilities to performing long-term follow-up measurements. Long-term (cohort) follow-up may also have practical constraints, such as participant burden and the transition of children from preschool to primary school, greatly affecting its feasibility. Researchers might have to critically consider whether weight-related outcomes should be included in relatively short-term effectiveness studies. 

### Strengths and Limitations

One strength of the current study is the comprehensive, integrated intervention approach, taking into account the complexity of childhood overweight and obesity. Therefore, children who may be at the highest risk for unhealthy lifestyles were exposed to the potential benefits of SuperFIT, as it was implemented in low-SES communities [75].

Other strengths included the objective assessment of BMI and PA outcomes. Changes in the preschool activities were assessed by observation and therefore evaluated more objectively compared to self-reporting. In addition, a long-term follow-up of approximately one and one-and-a-half years was used to evaluate effectiveness.

The methodology used may have some limitations. Not all participants adhered sufficiently to the accelerometer-wearing protocol to be included in the analyses (13.4% baseline, 12.2% first follow-up, and 9.1% final follow-up). Hip-worn accelerometers may overestimate sedentary behavior, as they do not measure posture and, for example, classify standing as sedentary [59,76]. Other methodological limitations may be that no blinding could be employed due to the nature of the study and its measurements. This may have induced reactivity to the measurements, for example increased physical activity because of the awareness of wearing the accelerometer, which may partially explain the changes seen in the control group.

In addition, it is possible that similar interventions were carried out in the control group. A large, national, community-based approach was also being implemented in the control group [77]. Although the projects were predominantly aimed at primary schools, it is likely that spill-over occurred as most preschools are situated inside the buildings of primary schools. The participating children may have had older school-going brothers or sisters who may have contributed to this spill-over. However, evaluation studies in real-life settings are important to determine intervention effectiveness, including all influential factors present in the real world, and decrease the gap in effects between efficacy and effectiveness studies [78].

Lastly, the sample size in this study was limited and, therefore, there was a lack of power to detect significant differences as the effect sizes were predominantly small or very small. To increase the power, analyses were performed comparing the complete intervention group with the control group. These analyses showed similar results and conclusions (data not shown).

## 5. Conclusions

Compared to the control group, no differences were detected between the groups to support the effectiveness of SuperFIT on weekly physical activity and weight. SuperFIT may nevertheless ultimately induce changes in the preschool setting regarding physical activity and sedentary behavior. The combination of the preschool setting and home setting appears to be beneficial for improving the children’s physical activity on preschool days. However, these effects are predominantly small and are not yet associated with changes in the BMI z-score.

## Figures and Tables

**Table 1 ijerph-16-05016-t001:** Baseline characteristics of the participants.

Characteristic	Full Intervention (N = 47)		Partial Intervention (N = 52)		Control (N = 92)	
	N (%) *	Mean ± SD	N (%) *	Mean ± SD	N (%) *	Mean ± SD
Child age (years)		3.2 ± 0.5		3.0 ± 0.4		3.1 ± 0.6
Child gender						
*Boy*	20 (42.6)		24 (46.2)		44 (47.8)	
*Girl*	27 (57.4)		28 (53.8)		48 (52.2)	
Child BMI z-score		0.25 ± 1.02		0.16 ± 0.88		0.13 ± 0.96
Parent age (years)		34.5 ± 4.1		35.6 ± 4.3		33.2 ± 4.3
Partner yes/no	26 (86.7)/4 (13.3)		35 (94.6)/2 (5.4)		37 (80.4)/9 (19.6)	
Parent birth country ^$^						
*Netherlands*	38 (92.7)		39 (90.7)		45 (68.2)	
*Other*	3 (7.3)		4 (9.3)		21 (31.8)	
Partner birth country						
*Netherlands*	33 (89.2)		34 (82.9)		43 (81.1)	
*Other*	4 (10.8)		7 (17.1)		10 (18.9)	
Parent education level ^$^						
*Low*	5 (12.2)		7 (16.3)		20 (30.8)	
*Middle*	12 (29.3)		15 (34.9)		27 (41.5)	
*High*	24 (58.5)		21 (48.8)		18 (27.7)	
Partner education level ^$^						
*Low*	5 (13.9)		5 (12.8)		18 (32.7)	
*Middle*	20 (55.6)		15 (38.5)		23 (41.8)	
*High*	11 (30.6)		19 (36.5)		14 (25.5)	
Parent BMI		24.7 ± 3.1		25.5 ± 3.5		24.4 ± 3.4

* Due to missing data N can vary, percentages are based on available data; ^$^ Significant difference between the groups. SD = standard deviation.

**Table 2 ijerph-16-05016-t002:** Changes in time spent in sedentary or physical activity (PA) activities at intervention preschools.

PA or Sedentary Activity		1st Obs.			2nd Obs.			3rd Obs.	
	Loc.	Min./Max.	Mean (SD)	Loc.	Min./Max.	Mean (SD)	Loc.	Min./Max.	Mean (SD)
Outside play (minutes)	7	10/28	20.1 (7.7)	7	10/33	23.1 (8.2)	7	13/60	29.1 (17.8)
Inside active play* (minutes)	8	3/24	11.4 (6.9)	7	6/28	18.0 (8.1)	9	1/69	18.3 (21.8)
Total active ** (minutes)	10	3/41	23.2 (14.9)	10	22/36	28.8 (4.7)	10	11/74	36.9 (20.9)
Inside sedentary (minutes)	9	13/68	33.8 (17.7)	8	15/47	29.1 (11.0)	10	5/45	22.5 (12.3)
Circle time (minutes)	9	11/47	22.9 (12.0)	9	8/33	18.8 (7.5)	9	12/29	19.8 (5.1)
Snack time (minutes)	10	20/32	25.8 (3.6)	10	15/33	24.2 (5.7)	10	15/31	23.5 (5.0)
Total sedentary ^$^ (minutes)	10	27/106	76.8 (25.7)	10	38/100	64.4 (21.1)	10	48/85	63.8 (13.0)

* Based on time spent in teacher-initiated activities ** Sum of outside play and inside active play ^$^ Sum of inside sedentary time, circle time, and snack time. PA = physical activity, Loc. = locations, obs. = observation, min. = minimum, max. = maximum, SD = standard deviation.

**Table 3 ijerph-16-05016-t003:** Effects of SuperFIT on physical activity outcomes on preschool days for each study group.

PA Outcome		Full Intervention *		Partial Intervention *		Control		Full vs. Control			Partial vs. Control		
		N	Mean ± SD **	N	Mean ± SD **	N	Mean ± SD **	B ^a^ (95% CI)	P	ES	B ^a^ (95% CI)	P	ES
SB (%)	T0	33	82.61 ± 5.44	41	81.57 ± 5.96	63	81.69 ± 5.50						
	T1	18	79.34 ± 6.34 ^$^	26	82.13 ± 4.62	46	81.83 ± 4.62	−3.99 (−7.11; −0.87)	0.01	−0.62	0.78 (−1.97; 3.53)	0.57	0.07
	T2	7	80.31 ± 3.90 ^$^	11	79.85 ± 6.55 ^$^	24	79.13 ± 4.94 ^$^	0.19 (−4.12; 4.50)	0.93	0.05	0.94 (−2.61; 4.50)	0.59	0.15
LPA (%)	T0	33	9.08 ± 2.10	41	9.45 ± 2.42	63	9.07 ± 2.33						
	T1	18	9.53 ± 2.27	26	9.24 ± 2.00	46	9.17 ± 1.97	0.38 (−0.90; 1.66)	0.56	0.16	−0.42 (−1.56; 0.72)	0.47	−0.13
	T2	7	10.81 ± 1.67 ^$^	11	9.72 ± 2.98 ^$^	24	10.47 ± 2.57 ^$^	0.38 (−1.82; 2.58)	0.73	0.15	−1.11 (−3.00; 0.76)	0.24	−0.48
MVPA (%)	T0	33	8.31 ± 3.57	41	8.98 ± 3.79	63	9.24 ± 3.60						
	T1	18	11.13 ± 4.54 ^$^	26	8.63 ± 2.95	46	9.00 ± 3.11	3.47 (1.39; 5.55)	0	0.85	−0.44 (−2.27; 1.39)	0.63	−0.03
	T2	7	8.89 ± 2.56 ^$^	11	10.43 ± 4.23 ^$^	24	10.40 ± 3.09 ^$^	−0.26 (−2.78; 2.25)	0.83	−0.16	−0.09 (−2.18; 1.99)	0.93	0.08
CPM	T0	33	1095.55 ± 315.17	41	1113.27 ± 274.91	63	1135.15 ± 296.62						
	T1	18	1197.48 ± 356.17	26	1095.27 ± 216.55	46	1100.77 ± 242.51	170.94 (8.86; 333.01)	0.04	0.45	−11.68 (−154.77; 131.41)	0.87	0.06
	T2	7	1230.54 ± 158.07 ^$^	11	1217.36 ± 361.45 ^$^	24	1281.17 ± 260.20 ^$^	24.01 (−218.55; 266.58)	0.84	−0.04	−69.05 (−268.17; 130.07	0.49	−0.15

* Full intervention = exposed to both preschool-based and family-based component, partial intervention = exposed to only preschool-based component ** Observed scores for each measurement ^a^ Linear mixed-model analysis, based on imputed estimated means, corrected for baseline, child age, child gender, and weather (temperature, precipitation, and sunshine), ^$^ Significantly different from baseline score, analyzed with paired *t*-tests. CI = confidence interval, CPM = counts per minute, ES = effect size, LPA = light physical activity, MVPA = moderate-to-vigorous physical activity, SB = sedentary behavior, SD = standard deviation.

**Table 4 ijerph-16-05016-t004:** Effects of SuperFIT on overall physical activity for each study group.

PA Outcome		Full Intervention *		Partial Intervention *		Control		Full vs. Control			Partial vs. Control		
		N	Mean ± SD **	N	Mean ± SD **	N	Mean ± SD **	B ^a^ (95% CI)	P	ES	B ^a^ (95% CI)	P	ES
SB (%)	T0	33	81.95 ± 5.30	41	81.34 ± 4.53	69	81.76 ± 5.58						
	T1	29	80.14 ± 6.39	31	80.00 ± 4.16	69	80.31 ± 5.31	0.65 (−1.66; 2.95)	0.58	−0.07	0.63 (−1.57; 2.83)	0.57	0.02
	T2	24	78.64 ± 4.20 ^$^	31	78.64 ± 4.59 ^$^	65	77.32 ± 5.42 ^$^	1.64 (−0.51; 3.79)	0.14	0.21	1.34 (−0.59; 3.27)	0.17	0.33
LPA (%)	T0	33	9.42 ± 2.04	41	9.80 ± 1.91	69	9.14 ± 2.25						
	T1	29	9.41 ± 2.49	31	9.96 ± 1.71	69	9.60 ± 2.13	−0.73 (−1.76; 0.30)	0.17	−0.22	−0.40 (−1.39; 0.59)	0.42	−0.14
	T2	24	10.50 ± 1.56	31	10.40 ± 2.23 ^$^	65	10.66 ± 1.73 ^$^	−0.66 (−1.61; 0.29)	0.17	0.01	−0.74 (−1.60; 0.12)	0.09	−0.22
MVPA (%)	T0	33	8.63 ± 3.46	41	8.86 ± 2.90	69	9.10 ± 3.71						
	T1	29	10.45 ± 4.20	31	10.05 ± 2.72	69	10.09 ± 3.56	0.17 (−1.28; 1.62)	0.82	0.23	−0.17 (−1.55; 1.21)	0.81	0.06
	T2	24	10.86 ± 3.67 ^$^	31	10.96 ± 2.94 ^$^	65	12.02 ± 4.22 ^$^	−0.81 (−2.27; 0.65)	0.27	−0.19	−0.65 (−1.96; 0.66)	0.33	−0.24
CPM	T0	33	1111.23 ± 271.86	41	1138.15 ± 230.96	69	1132.32 ± 313.80						
	T1	29	1143.83 ± 348.79	31	1189.96 ± 201.46	69	1172.41 ± 284.56	−68.84 (−194.65; 56.99)	0.28	−0.02	−29.92 (−149.85; 90.02)	0.62	0.04
	T2	24	1300.09 ± 248.31 ^$^	31	1297.31 ± 241.94 ^$^	65	1383.19 ± 328.47 ^$^	−95.96 (−225.88; 33.95)	0.15	−0.21	−81.14 (−197.96; 35.67)	0.17	−0.32

* Full intervention = exposed to both preschool-based and family-based component, partial intervention = exposed to only preschool-based component ** Observed scores for each measurement ^a^ Linear mixed-model analysis, based on imputed estimated means, corrected for baseline, child age, child gender, and weather (temperature, precipitation, and sunshine) ^$^ Significantly different from baseline score, analyzed with paired *t*-tests. CI = confidence interval, CPM = counts per minute, ES = effect size, LPA = light physical activity, MVPA = moderate-to-vigorous physical activity, SB = sedentary behavior, SD = standard deviation.

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
