# Peer review of "The Effects of a Comprehensive, Integrated Obesity Prevention Intervention Approach (SuperFIT) on Children’s Physical Activity, Sedentary Behavior, and BMI Z-Score"

_ijerph, 2019, doi:10.3390/ijerph16245016_

Round 1
Reviewer 1 Report
The authors examined the effects of a multi-level obesity prevention intervention on young children’s physical activity, sedentary behavior, and BMI z-score. Children aged 2-4 years wore accelerometers and were observed during pre-school activities. The intervention took place in the preschools, while a subset of the children participated in a family component of the intervention. A control group was recruited from similar preschools in the area. Studies like the one presented that include interventions targeting obesity prevention among young children are essential for reducing the onset of obesity. Although the study design integrates multiple levels of the socioecological model, presentation of these separated outcomes (preschool only vs preschool plus family) may not be warranted.
The following comments are general regarding the manuscript:
Data are plural. Please adjust this throughout your manuscript. Because the referenced protocol is not currently in publication and reviewers are unable to review it, the authors should ensure that enough detail is provided regarding the study design and methodology. For example, how many sessions did caregivers attend? Few details are provided.Additional comments are specific to sections or lines of the manuscript:
Title and Abstract: Sedentary time is included as an outcome in the text of the manuscript (Line 71), but is not included in the abstract as a study aim. It is also not included in the title.
Line 90: The authors indicate that intervention strategies were designed to be add-in rather than add-on. Based on additional descriptions (line 99), are these strategies meant to be integrated into the existing curriculum? If so, this description would be welcomed earlier (with the information provided in Line 90).
Sec 2.3: For children in intervention schools who did not participate, did the teachers separately instruct the children who did not agree to the study? If teachers did not agree to participate, were separate teachers brought in to the classrooms? It is not clear what happened to the children and teachers who were not participating.
Sec 2.4.2: Observations at the preschool are included in Table 2, but no statistical examinations were conducted to examine the differences between them. Is there a reason for not statistically examining changes over time?
Line 209: The authors note use of standard errors to calculate effect sizes. Should this read standard deviations instead?
Line 370: The authors indicate that they collapsed the data from the two intervention groups, with similar results as those presented. In essence, this means that the family component had no greater influence on the results presented (BMI z-scores and physical activity). The authors do not provide details about the parenting component of the intervention. Since this is the case, and the authors did not examine parental mediators of the SuperFIT program on the outcomes, why not present the collapsed data rather than separating out the two intervention components?
Author Response
The authors examined the effects of a multi-level obesity prevention intervention on young children’s physical activity, sedentary behavior, and BMI z-score. Children aged 2-4 years wore accelerometers and were observed during pre-school activities. The intervention took place in the preschools, while a subset of the children participated in a family component of the intervention. A control group was recruited from similar preschools in the area. Studies like the one presented that include interventions targeting obesity prevention among young children are essential for reducing the onset of obesity. Although the study design integrates multiple levels of the socioecological model, presentation of these separated outcomes (preschool only vs preschool plus family) may not be warranted.
Author’s response:
Thank you for your review and suggestions to improve the manuscript. We will address the issue on the study design at the specific question that you asked about it.
The following comments are general regarding the manuscript:
Data are plural. Please adjust this throughout your manuscript.
Author’s response:
Thank you for noticing. We have adjusted the manuscript throughout.
Because the referenced protocol is not currently in publication and reviewers are unable to review it, the authors should ensure that enough detail is provided regarding the study design and methodology. For example, how many sessions did caregivers attend? Few details are provided.
Author’s response:
We agree with the reviewer that all necessary information regarding the study design and methodology should be published. As the protocol is currently not published yet, we have provided more detail on the study design and methodology. Taking into account your comment, the study protocol is now submitted to this journal to be considered for publication in combination with this paper.
Correction:
Methods, section 2.2
Line 97-136: SuperFIT is a comprehensive, integrated intervention approach based on socio-ecological models of behaviour and systems theory [25,27]. In particular, it aims to align intervention strategies between different micro-settings (i.e. preschool and home) to enhance intervention effectiveness. SuperFIT was developed in partnership with the intervention preschool organization, a local PA providing organization, and health promotion experts. Formative research [35], practice-based knowledge of cooperating partners, and theory- and evidence-based knowledge [26,36-39] were used to develop it. Further, a continuous process of co-creation, feedback and adaptations was adopted during development to increase suitability and applicability in the settings. The intervention consisted of preschool-based, family-based, and community components.
Within the preschool-based component, several strategies were implemented that targeted the sociocultural and physical environment [40]. The sociocultural environment was operationalised as the physical activity- and nutrition-related practices of the preschool teachers. The intervention strategies involved the preschool teachers and included 1) an inspirational session with a school-based PA expert, 2) three off-the-job interactive training sessions on PA, nutrition, and positive child rearing, led by an expert on the specific topic 3) on-the-job coaching sessions following each the off-the-job training (three in total), on PA or nutrition led by the same experts, and 4) cards with easy-to-perform PA games and nutrition-related activities to support preschool teachers to integrate PA and healthy nutrition in the curriculum. The physical environment was operationalised as all that is tangibly available for the children at the preschool. Intervention strategies were 1) provision of a box of general play materials aligned with the PA-related cards that could promote PA both indoors and outdoors (e.g. bean bags, hoops, balls), 2) general nutrition-related materials aligned with the nutrition-related cards (e.g. water tap, fruit and vegetable toys, nutrition-related story books), and 3) complementary fruit and vegetables delivery. A local greengrocer supplied less familiar fruits or vegetables (e.g. avocado, raspberries, carrots in different colours) to increase variety. These supplemental fruit and vegetables were available every day and were similar every two weeks to ensure repeated exposure. Intervention strategies were mostly designed to serve as add-in rather than add-on activities, which were highly adaptable to the specific situation of a preschool, to support implementation and sustainability.
The family-based component was developed to provide fun activities to help families integrate PA and healthy nutrition in their daily life [41]. All possible caregivers were invited to participate in these sessions. Caregiver-only sessions were organized in order to address the influences of the different types of environments (i.e. sociocultural, physical, political and economic [40]) on nutrition and PA. These sessions were based on Lifestyle Triple P seminars and were given by a trained Triple P implementer [42]. Three Triple P seminars were provided (one on PA, one on nutrition and one on general parenting), which lasted around 1.5 hours. During the caregiver-only sessions PA or nutrition-related activities were organised for the children and childcare was available for the youngest siblings. In addition, family sessions were organized. They were characterised by fun activities for the whole family. The PA-related family sessions aimed at co-physical activity, performing active games that are easily translated to the home setting. The nutrition-related family sessions consisted of, for example, taste sessions and making healthy treats.
Additional comments are specific to sections or lines of the manuscript:
Title and Abstract: Sedentary time is included as an outcome in the text of the manuscript (Line 71), but is not included in the abstract as a study aim. It is also not included in the title.
Author’s response:
We have now added sedentary behaviour to the title and abstract as one of the outcomes in the manuscript.
Corrections:
Title
The effects of a comprehensive, integrated obesity prevention intervention approach (SuperFIT) on children’s physical activity, sedentary behaviour and BMI z-score.
Abstract
Line 16-17: A quasi-experimental research design was adopted to evaluate the effects of SuperFIT on physical activity (PA), sedentary behaviour (SB) and BMI z-score.
Line 19-24: Children’s PA levels and SB were assessed with accelerometers and observations, and height and weight were measured for the BMI z-score. Measurements were performed at baseline and two follow-up time points. Effectiveness was evaluated using linear mixed-model analyses, correcting for relevant covariates. Healthy changes in PA levels occurred within all study groups over time. No significant differences were found in overall PA levels between the intervention groups and control group at both follow-ups.
Line 90: The authors indicate that intervention strategies were designed to be add-in rather than add-on. Based on additional descriptions (line 99), are these strategies meant to be integrated into the existing curriculum? If so, this description would be welcomed earlier (with the information provided in Line 90).
Author’s response:
The reviewer is right that intervention strategies were indeed aimed to be integrated into the existing curriculum. Moreover, the strategies were intended to fit as much as possible to existing practices at the preschools. For example, most preschools have a scheduled time for active play. Through the provision of the play materials and the training and coaching of the preschool teachers, the intervention strategies aimed to support preschool teachers to stimulate physical activity of the children during these scheduled active play moments. Therefore, no new activities were introduced that had to be added to the daily schedule, but activities would fit the existing schedule. We have adjusted the order of this part of the intervention description as suggested by the reviewer.
Corrections:
Methods
Line 107-124: Within the preschool-based component, several strategies were implemented that targeted the sociocultural and physical environment [41]. The sociocultural environment was operationalised as the physical activity- and nutrition-related practices of the preschool teachers. The intervention strategies involved the preschool teachers and included 1) an inspirational session with a school-based PA expert, 2) three off-the-job interactive training sessions on PA, nutrition, and positive child rearing, led by an expert on the specific topic 3) on-the-job coaching sessions following each the off-the-job training (three in total), on PA or nutrition led by the same experts, and 4) cards with easy-to-perform PA games and nutrition-related activities to support preschool teachers to integrate PA and healthy nutrition in the curriculum. The physical environment was operationalised as all that is tangibly available for the children at the preschool. Intervention strategies were 1) provision of a box of general play materials aligned with the PA-related cards that could promote PA both indoors and outdoors (e.g. bean bags, hoops, balls), 2) general nutrition-related materials aligned with the nutrition-related cards (e.g. water tap, fruit and vegetable toys, nutrition-related story books), and 3) complementary fruit and vegetables delivery. A local greengrocer supplied less familiar fruits or vegetables (e.g. avocado, raspberries, carrots in different colours) to increase variety. These supplemental fruit and vegetables were available every day and were similar every two weeks to ensure repeated exposure. Intervention strategies were mostly designed to serve as add-in rather than add-on activities, which were highly adaptable to the specific situation of a preschool, to support implementation and sustainability.
Sec 2.3: For children in intervention schools who did not participate, did the teachers separately instruct the children who did not agree to the study? If teachers did not agree to participate, were separate teachers brought in to the classrooms? It is not clear what happened to the children and teachers who were not participating.
Author’s response:
All children attending the preschools were exposed to SuperFIT, as the childcare organisation decided to implement the intervention as part of the curriculum. Parents could give consent to participate in the research concerning SuperFIT. There was no difference in the daily practice of the preschool teachers for children whose parents agreed to participate in the research and for children whose parents did not agree.
With regard to the preschool teachers, they also provided informed consent to participate in the research concerning SuperFIT. The intervention itself was seen as part of their professional development. The childcare organization expected all preschool teachers to participate in the training and coaching sessions provided within SuperFIT.
We have clarified this difference in section 2.3.
Corrections:
Methods, section 2.3
Line 157-160: All children attending the participating preschools were exposed to SuperFIT, although not all children participated in the research. All preschool teachers were expected to participate in the intervention training and coaching sessions by the childcare organization as part of their professional development.
Sec 2.4.2: Observations at the preschool are included in Table 2, but no statistical examinations were conducted to examine the differences between them. Is there a reason for not statistically examining changes over time?
Author’s response:
The number of observations is limited, as they were performed on the preschool level (N=10). This does not provide sufficient power for statistical analysis. Therefore, we decided to show only results of descriptive analysis of the observations. The goal of the observations was not to study effects, but observe changes in the preschool setting.
Line 209: The authors note use of standard errors to calculate effect sizes. Should this read standard deviations instead?
Author’s response:
The effect sizes were calculated based on the estimated means derived from the mixed model analyses. These are provided with a standard error, as they are no longer derived from observed data. The standard error was used to calculate the effect size. To prevent confusion, we think it is appropriate to report the use of the standard error.
Line 370: The authors indicate that they collapsed the data from the two intervention groups, with similar results as those presented. In essence, this means that the family component had no greater influence on the results presented (BMI z-scores and physical activity). The authors do not provide details about the parenting component of the intervention. Since this is the case, and the authors did not examine parental mediators of the SuperFIT program on the outcomes, why not present the collapsed data rather than separating out the two intervention components?
Author’s response:
First, we have provided increased detail on the parenting component in the Methods section.
Second, we decided to show the results separately for the two intervention groups for several reasons. The first reason is the design of the study. As the children in the family component were exposed to the complete SuperFIT intervention, it would not be appropriate to combine them with the children that were only exposed to the preschool component (partial intervention). This would result in an intervention group of participants with different exposures and therefore, it would remain unclear to what exposure the effects are relating.
The second reason is that there are differences in the effects of SuperFIT for children participating in the full intervention and those participating in the partial intervention. Most of these differences are favourable for the full intervention group. This indicates that there is a beneficial effect of the family component. This difference would not have been visible if presenting the combined data. In addition, we did analyse the collapsed data because this would increase statistical power, and might show trends that were not visible in the separate group analysis. However, as mentioned in the manuscript, this increase in power did not change the results or conclusions about SuperFIT.
Reviewer 2 Report
This paper reports on a comprehensive, integrated intervention in childcare settings in Netherland. This paper contains some information about the various approaches within the overall intervention. I have several suggestions to help the authors improve the quality of the paper.
Introduction.
1.This section is rather brief and doesn't provide a complete rationale to the reported intervention. Please strengthen the rationale of conducting this study in Netherland? Was it just a duplicated study in Netherland?
2.The hypothesis of the manuscript are not well presented. Please provide the hypothesis with clear sentences, specify the hypothesis at the end of the Introduction section.
3.Line 66-67, the author declared that “It primarily aims to change the energy balance-related behaviours of children”. It is my opinion that the author should report the outcome of the changes in energy balance-related behaviours.
Method
4.More details of the protocol of the SuperFIT intervention should be here or had been published.
5.No information is provided on the method of preschool recruitment..
6.The information on who developed and delivered certain training sessions should be included to give other practitioners or public health researchers actually information on the practical runnings of the intervention and training components.
7.In the Childcare Centre intervention paragraph, who delivered the teacher training sessions and what were their qualifications? Who developed the curriculum and what were their qualifications? How was quality of implementation monitored?
8.Needs to be clear whether the control preschool /children in the control preschool was located near to the community intervention neighbourhood. There is the potential for contamination if the control children had accessed to the intervention community.
9.What was the study powered to detect a difference of? A post-hoc power calculation is necessary.
Analysis
10.Analysis would need to control for grade level (as there is a sig difference) and sex.
11.The family intervention data should be included.
12.Process information related to implementation of PA components of SuperFIT based on accelerometry should be more details,e.g. the time in LPA, MVPA,SB per day during weekdays, weekend, schooltime at weekdays.
Discussion
Why did the author’s participants see increase in BMI while other studies showed decrease with similar intervention? I also have some concerns relating to the environmental components of the program. There is no discussion on the use of these elements or their relationship to the results.Author Response
This paper reports on a comprehensive, integrated intervention in childcare settings in Netherland. This paper contains some information about the various approaches within the overall intervention. I have several suggestions to help the authors improve the quality of the paper.
Author’s response:
Thank you for your review and suggestions to improve the manuscript.
Introduction.
1.This section is rather brief and doesn't provide a complete rationale to the reported intervention. Please strengthen the rationale of conducting this study in Netherland? Was it just a duplicated study in Netherland?
Author’s response:
We thank the reviewer for this suggestion. This study was not a replication study. Based on the reviewer’s suggestion, we have extended the introduction. We have given specific attention to the rationale for conducting the intervention in the Netherlands.
Corrections:
Introduction
Line 41-42: In the Netherlands, around 8% of 2-year-old children are overweight or obese and this prevalence increases with age [13].
Line 67-70: Interventions tend to focus on single settings or single aspects of childhood overweight and obesity and as a result may lack comprehensiveness. Therefore, SuperFIT was developed as a comprehensive, integrated intervention approach in the ECE and home settings in the Netherlands. It takes into account the interaction between settings and the complexity of the childhood overweight and obesity problem
2.The hypothesis of the manuscript are not well presented. Please provide the hypothesis with clear sentences, specify the hypothesis at the end of the Introduction section.
Author’s response:
We thank the reviewer for pointing this out, and have provided the hypothesis of the study more clearly at the end of the introduction.
Corrections:
Introduction
Line 77-81: We hypothesize that SuperFIT will increase the levels of physical activity and decrease the levels of sedentary behaviour in young children compared to the control group. In addition, we hypothesize that SuperFIT will help children maintain or achieve a healthy BMI z-score. Finally, we hypothesize that the combination of the preschool and home setting will lead to synergy, increasing intervention effectiveness.
3.Line 66-67, the author declared that “It primarily aims to change the energy balance-related behaviours of children”. It is my opinion that the author should report the outcome of the changes in energy balance-related behaviours.
Author’s response:
Based on this suggestion, we have made the goals and outcomes more specific.
Corrections:
Introduction
Line 71-73: It is a comprehensive, integrated intervention approach in the ECE and home settings. It primarily aims to increase physical activity levels, decrease sedentary behaviour and increase healthy nutrition behaviour (e.g. drinking water and eating fruits and vegetables) of young children through changes in …
Method
4.More details of the protocol of the SuperFIT intervention should be here or had been published.
Author’s response:
We agree with the reviewer that all necessary information regarding the study design and methodology should be published. As the protocol is currently not published yet, we have provided more detail on the study design and methodology. Taking into account your comment, the study protocol is now submitted to this journal to be considered for publication in combination with this paper. This also covers comment 6 and 7, the change in the manuscript are described at comment 7.
5.No information is provided on the method of preschool recruitment..
Author’s reponse:
We understand that our description of the recruitment in section 2.3 is not yet sufficient. The preschool organization was part of the project from the beginning, and all participating preschools were part of this organization. We have added this in the intervention description in Methods, section 2.2. Further, we have added a sentence to Methods, section 2.3 to clarify how the final selection of intervention preschool was determined.
Corrections:
Methods, section 2.1
Line 99-101: SuperFIT was developed in partnership with the intervention preschool organization, a local PA providing organization, and health promotion experts.
Methods, section 2.2
Line 147-148: In consultation with the childcare management a selection of eligible preschools was determined.
6.The information on who developed and delivered certain training sessions should be included to give other practitioners or public health researchers actually information on the practical runnings of the intervention and training components.
7.In the Childcare Centre intervention paragraph, who delivered the teacher training sessions and what were their qualifications? Who developed the curriculum and what were their qualifications? How was quality of implementation monitored?
Author’s response:
As described at comment 5, we agree with the reviewer that all necessary information regarding the study design and methodology should be published. Based on the reviewers comments (6 and 7) we have provided more detail on the intervention description to make it more clear what SuperFIT practically looked like.
Correction:
Methods, Section 2.2
Lines 97-136: SuperFIT is a comprehensive, integrated intervention approach based on socio-ecological models of behaviour and systems theory [25,27]. In particular, it aims to align intervention strategies between different micro-settings (i.e. preschool and home) to enhance intervention effectiveness. SuperFIT was developed in partnership with the intervention preschool organization, a local PA providing organization, and health promotion experts. Formative research [35], practice-based knowledge of cooperating partners, and theory- and evidence-based knowledge [26,36-39] were used to develop it. Further, a continuous process of co-creation, feedback and adaptations was adopted during development to increase suitability and applicability in the settings. The intervention consisted of preschool-based, family-based, and community components.
Within the preschool-based component, several strategies were implemented that targeted the sociocultural and physical environment [40]. The sociocultural environment was operationalised as the physical activity- and nutrition-related practices of the preschool teachers. The intervention strategies involved the preschool teachers and included 1) an inspirational session with a school-based PA expert, 2) three off-the-job interactive training sessions on PA, nutrition, and positive child rearing, led by an expert on the specific topic 3) on-the-job coaching sessions following each the off-the-job training (three in total), on PA or nutrition led by the same experts, and 4) cards with easy-to-perform PA games and nutrition-related activities to support preschool teachers to integrate PA and healthy nutrition in the curriculum. The physical environment was operationalised as all that is tangibly available for the children at the preschool. Intervention strategies were 1) provision of a box of general play materials aligned with the PA-related cards that could promote PA both indoors and outdoors (e.g. bean bags, hoops, balls), 2) general nutrition-related materials aligned with the nutrition-related cards (e.g. water tap, fruit and vegetable toys, nutrition-related story books), and 3) complementary fruit and vegetables delivery. A local greengrocer supplied less familiar fruits or vegetables (e.g. avocado, raspberries, carrots in different colours) to increase variety. These supplemental fruit and vegetables were available every day and were similar every two weeks to ensure repeated exposure. Intervention strategies were mostly designed to serve as add-in rather than add-on activities, which were highly adaptable to the specific situation of a preschool, to support implementation and sustainability.
The family-based component was developed to provide fun activities to help families integrate PA and healthy nutrition in their daily life [41]. All possible caregivers were invited to participate in these sessions. Caregiver-only sessions were organized in order to address the influences of the different types of environments (i.e. sociocultural, physical, political and economic [40]) on nutrition and PA. These sessions were based on Lifestyle Triple P seminars and were given by a trained Triple P implementer [42]. Three Triple P seminars were provided (one on PA, one on nutrition and one on general parenting), which lasted around 1.5 hours. During the caregiver-only sessions PA or nutrition-related activities were organised for the children and childcare was available for the youngest siblings. In addition, family sessions were organized. They were characterised by fun activities for the whole family. The PA-related family sessions aimed at co-physical activity, performing active games that are easily translated to the home setting. The nutrition-related family sessions consisted of, for example, taste sessions and making healthy treats.
8.Needs to be clear whether the control preschool /children in the control preschool was located near to the community intervention neighbourhood. There is the potential for contamination if the control children had accessed to the intervention community.
Author’s response:
The control preschools were at sufficient distance to the intervention preschools (on average 32 km apart) to prevent contamination of the intervention into the control region. The children, parents and preschool teachers in the control group did not have access to the intervention. However, as described in the discussion (line 399-404), it is possible that contamination occurred due to other national, regional or community initiatives to increase physical activity in young children.
9.What was the study powered to detect a difference of? A post-hoc power calculation is necessary.
Author’s response:
Based on the sample size of the study we were able to detect an increase of 1.44% MVPA and a difference of 0.19 BMI z-points. The data shows that these changes were not achieved, as observed differences were smaller. As explained by Hoenig and Heisey (2001), post hoc power is already reflected in the p-value and confidence interval; non-significant p values correspond to low post hoc powers [1]. Further, scholars agree that post hoc power does not add new information to p-values and confidence intervals [1,2]. Although, we indicated which results were statistically significant and which were not, we did not include the corresponding p-values. We have now added these to the Tables and supplementary materials.
Hoenig, J.M.; Heisey, D.M. The Abuse of Power. The American Statistician 2001, 55, 19-24, doi:10.1198/000313001300339897. Lenth, R. Post Hoc Power: Tables and Commentary. 2007.
Corrections:
Table 3 & 4: p-values are added for the outcomes of the mixed model analyses.
Supplementary material 1 & 2: p-values are added for the outcomes of the mixed model analyses.
Results
Line 303-312: No significant differences in BMI z-score were seen, with very small effect sizes between the full intervention and control group at the first follow-up (observed mean±SD 0.20±0.98 and 0.13±1.00 resp.; B= -0.09, 95%CI -0.31; 0.13, p= 0.44, ES -0.09) and the final follow-up (observed mean±SD 0.28±0.90 and 0.08±0.94 resp.; B= 0.00, 95%CI -0.25; 0.25, p= 0.99 ES 0.01). In addition, no significant differences were seen between the partial intervention and control group on BMI z-score at the first follow-up (observed mean±SD 0.20±0.81 and 0.13±1.00 resp.; B= 0.05, 95%CI -0.17; 0.26, p=0.66, ES 0.06) and the final follow-up (observed mean±SD -0.01±0.77 and 0.08±0.90 resp.; B= -0.13, 95%CI -0.38; 0.11, p= 0.28, ES -0.14).
Analysis
10.Analysis would need to control for grade level (as there is a sig difference) and sex.
Author’s response:
The analyses presented in the manuscript are corrected for sex of the children (described in Methods, section 2.4.4). In the Netherlands, preschool does not consist of grades. There is only one group. Therefore, additional correction for grade level is not possible nor necessary (described in Methods, section 2.5).
11.The family intervention data should be included.
Author’s response:
Data of the family component are presented in both Table 3 and 4. Participants of the family component were also exposed to the preschool component. Therefore, they are described as the full intervention group. This is also described in Methods, section 2.3, lines 154-156.
12.Process information related to implementation of PA components of SuperFIT based on accelerometry should be more details,e.g. the time in LPA, MVPA,SB per day during weekdays, weekend, schooltime at weekdays.
Author’s response:
Process information regarding SuperFIT will be evaluated separately.
Children in the full intervention spent 498 minutes in SB, 58 minutes in LPA, and 53 minutes in MVPA at baseline. This increased to 530 minutes SB, 71 minutes LPA, and 73 minutes MVPA at final follow-up. In the partial intervention group, children spent 499 minutes in SB, 60 minutes in LPA, and 54 minutes in MVPA at baseline. This increased to 499 minutes in SB, 67 minutes in LPA, and 70 minutes in MVPA at final follow-up. Children in the control group spent 515 minutes in SB, 58 minutes in LPA, and 58 minutes in MVPA at baseline. They decreased SB to 480 minutes and increase LPA to 66 minutes and MVPA to 74 minutes. We chose not to report the time in SB, LPA and MVPA because this is dependent on the amount of wear time of the accelerometer. A relative value (percentage) of these physical activity levels is therefore reported, correcting for wear time. As SuperFIT aimed to increase overall physical activity levels, we chose to report on weekly PA levels and not separately for week and weekend days. We did choose to analyse preschool days separately, because the preschool component of SuperFIT specifically targeted PA at preschool. As we did not collect information regarding specific times the child was present at the preschool, we were not able to analyze effects during preschool time only.
Discussion
Why did the author’s participants see increase in BMI while other studies showed decrease with similar intervention? I also have some concerns relating to the environmental components of the program. There is no discussion on the use of these elements or their relationship to the results.
Author’s response:
With regard to BMI z-score, as explained in the discussion BMI z-score was the most distal outcome of our study. We expected changes in the physical and social environment would lead to changes in child behaviours, which would lead to changes in BMI z-score. This is a long causal chain. Therefore, the small behavioural effects in this study may not have been substantial enough, or the follow-up time may not be long enough, to also cause changes in BMI z-score (line 328 – 333). The children in this study may have continued on the normal expected growth, in which the number of children with overweight or obesity increase with increasing age. This may explain the small (but non-significant) increases in BMI z-score in some subsamples.
We have added a paragraph to the discussion to explain the influence of the environmental components of SuperFIT on the results.
Corrections:
Discussion
Line 358-368: Studies have shown the importance of determinants in different types of environments on physical activity and sedentary behaviour [35, 60, 61]. Systematic reviews have also stressed the importance of integrated interventions, i.e. combining the physical and sociocultural environment [62, 63]. The integrated approach taken in SuperFIT, in which these different types of environments were taken into account was expected to be supportive for intervention effectiveness. The strategies that were implemented to change the different environments were evidence-based [42, 64-66]. In addition, a review has shown that environmental changes are most effective to stimulate pre-schoolers physical activity [24]. Due to the comprehensive, integrated approach of SuperFIT, it is difficult to identify which elements supported effectiveness. To increase our understanding, separate evaluations will be performed to study the effects of SuperFIT on the physical and social environment.
Round 2
Reviewer 2 Report
It is ok.